# CORRECTNESS WITHOUT CONCEPTS? EVALUATING LLM REASONING BEYOND ANTHROPOCENTRIC TAXONOMIES

## ABSTRACT

Accuracy is the standard metric for evaluating LLM reasoning, but it conflates two distinct capabilities: understanding the underlying concept and executing it correctly. We introduce a two-phase evaluation framework that separates these concerns. A solver model attempts ConceptARC tasks requiring inductive reasoning from examples. A separate judge model evaluates only the reasoning trace, scoring conceptual understanding independent of output correctness. Across 480 evaluations (160 tasks $\times$ 3 passes), we find 38% show a mismatch: correct answers from flawed reasoning, or incorrect answers despite sound understanding. We analyze failure patterns across concept types, finding systematic weaknesses in spatial reasoning (Cohen's $d = 1.53$) and 34% inconsistency across repeated attempts. Finally, we note a key caveat: the concepts used in benchmarks like ConceptARC are human-defined and anthropocentric, while the internal abstractions LLMs use to reason may be very different. This motivates interpreting "concept understanding" scores as alignment with benchmark taxonomies, rather than a universal measure of conceptual structure.

## 1 INTRODUCTION

When an LLM solves a reasoning problem correctly, did it actually understand the underlying logic? Standard benchmarks report accuracy, treating correct outputs as evidence of reasoning capability. But a model can arrive at correct answers through pattern matching, memorization, or lucky guesses, without genuine conceptual understanding. Conversely, a model might understand a concept perfectly but make execution errors that produce incorrect outputs.

If we cannot distinguish understanding from correctness, we cannot diagnose *why* models fail, design targeted improvements, or trust models in high-stakes applications where reasoning matters.

We introduce a two-phase evaluation framework that measures understanding and correctness independently. Applied to ConceptARC (Moskvichev et al., 2023), a benchmark requiring induction of transformation rules from examples, we find that 38% of evaluations show a mismatch between understanding and correctness. This has significant implications: accuracy alone would misclassify more than one-third of cases.

More broadly, this framing raises a conceptual caveat: the "concepts" defined by ConceptARC reflect human-designed categories. It is plausible that LLMs may internally organize solutions around different, potentially non-anthropocentric abstractions, while still producing correct transformations. Our evaluation therefore measures alignment with ConceptARC's concept definitions, not a universal notion of what constitutes a "concept" for an LLM.

Our contributions are: (1) a two-phase evaluation methodology separating reasoning quality from output correctness; (2) evidence that 38% of evaluations show understanding-correctness mismatch; and (3) analysis of systematic failure patterns, including spatial reasoning weakness ($d = 1.53$), 34% inconsistency across attempts, and longer reasoning traces correlating with failure.

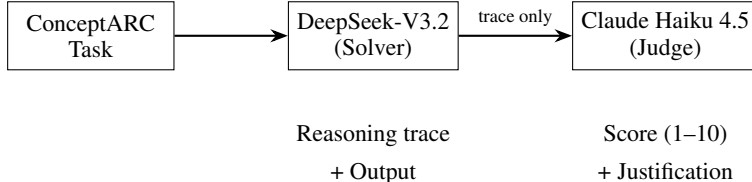

Figure 1: Two-phase evaluation pipeline. The solver generates reasoning traces and outputs. The judge evaluates only the reasoning trace, blind to output correctness, producing a score and justification.

## 2 METHODOLOGY

### 2.1 TASK: CONCEPTARC

ConceptARC (Moskvichev et al., 2023) is a benchmark for abstract visual reasoning organized around 16 concepts (e.g., Copy, Center, InsideOutside). Each task provides 2–3 input-output grid examples demonstrating a transformation rule. The model must induce the rule and apply it to new test inputs. We use 160 tasks (10 per concept), requiring inductive reasoning and generalization to novel inputs. Appendix **??** lists all concepts with their descriptions.

### 2.2 TWO-PHASE EVALUATION PIPELINE

Our evaluation separates reasoning quality from output correctness through two phases (Figure 1).

**Phase 1: Solver.** DeepSeek-V3.2 in Thinking Mode (DeepSeek-AI et al., 2025) generates predictions with extended reasoning traces (chain-of-thought). We used `max_tokens=64000` because initial runs with the default 32K limit resulted in 13 truncated responses (2.7%) across 10 tasks and 6 concepts, with the CleanUp concept most affected.

**Phase 2: Judge.** Claude Haiku 4.5 reads only the reasoning trace and concept name, not the output or whether it was correct. It produces a score (1–10) for conceptual understanding and a short justification explaining the rating.

This separation allows us to detect correct answers from flawed reasoning, and incorrect answers despite sound reasoning.

### 2.3 SCORING PROMPT DEVELOPMENT

We iteratively refined the scoring rubric to distinguish genuine conceptual understanding from mechanical pattern descriptions. The final rubric scores four independent dimensions: (1) rule-concept alignment (whether the derived rule matches the concept), (2) concept articulation (whether the model explicitly names or describes the concept), (3) application correctness (whether the rule is applied systematically), and (4) explanatory depth (whether the model explains *why* the rule works, not just what it does). The full prompts are provided in Appendix **??**.

Scores converge to 7–9 for most evaluations, which is expected: the model grasps most concepts adequately. Lower scores (1–4) indicate clear conceptual failures. Perfect scores (10) are rare because flawless articulation is uncommon even for correct solutions. Figure 2 shows the score distribution, with a strong peak at 8 (47%) and a secondary cluster at 2–3 (26%) representing cases of poor reasoning. This bimodal pattern suggests the rubric successfully discriminates between adequate and inadequate understanding.

### 2.4 EVALUATION PROTOCOL

Each task was evaluated 3 times to reduce sampling variance. We define "high understanding" as score $\geq 5$, separating low understanding (1–4) from adequate understanding (5+). Total evaluations: 480 (160 tasks $\times$ 3 passes). Statistical comparisons use Welch's $t$-test and Cohen's $d$ for effect size.

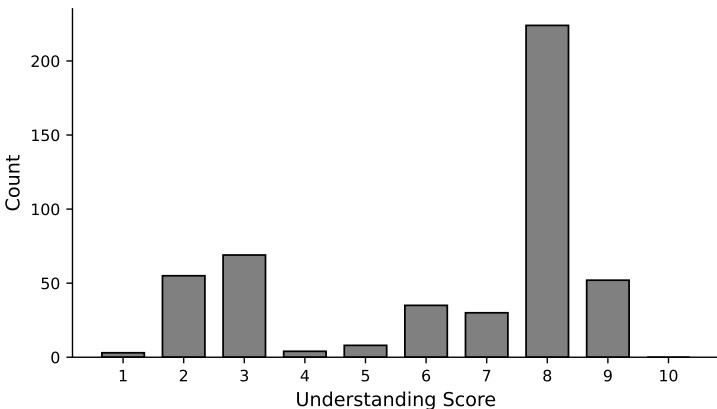

Figure 2: Distribution of understanding scores. Scores cluster at 8 (47%), with scores 2–3 well-populated for poor reasoning, indicating the rubric successfully discriminates.

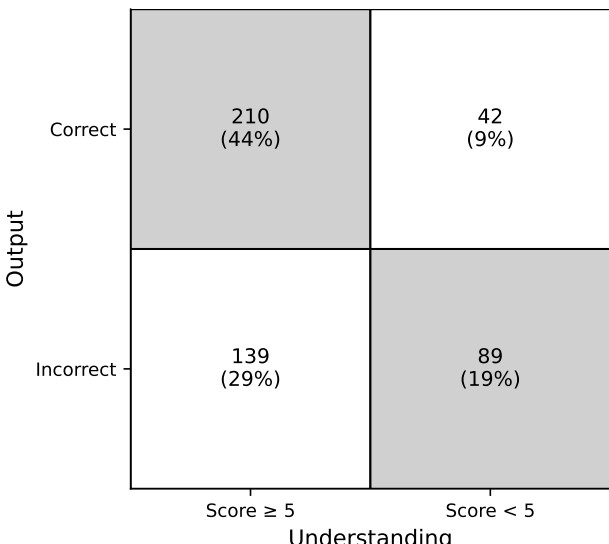

Figure 3: Understanding vs. correctness. 38% of evaluations show a mismatch: correct answers with low understanding (9%) or incorrect answers with high understanding (29%).

## 3 RESULTS

### 3.1 THE UNDERSTANDING-CORRECTNESS GAP

Figure 3 shows our central finding. We partition evaluations by correctness (correct/incorrect output) and understanding (score $\geq 5$ or $< 5$).

The overall accuracy is 52.5%, with a mean understanding score of 6.37. The correlation between understanding score and correctness is weak ($r = 0.24$), suggesting these measure partially distinct capabilities. Most importantly, the mismatch rate is 38%: more than one-third of evaluations show a disconnect between understanding and correctness. The 9% of cases with correct answers but low understanding represent potential "lucky guesses" or pattern matching without genuine comprehension. The 29% with incorrect answers but high understanding represent execution failures despite sound reasoning. Accuracy alone would misclassify both categories.

### 3.1.1 QUADRANT EXAMPLES

To illustrate what each quadrant represents, we show the judge's justification for representative cases.

**Correct + High Understanding** (ExtractObjects5, score 9): *"The derived rule precisely captures the core mechanism: systematically identify maximal hollow squares, extract them as independent objects, and arrange them horizontally by column position. This directly implements the concept of identifying and isolating distinct objects."*

**Correct + Low Understanding** (TopBottom2D8, score 1): *"The model completely failed to understand the TopBottom2D concept. The reasoning derives a rule about finding the last non-zero cell and recoloring, entirely unrelated to how shapes interact or hide each other on a 2D grid."* This case produced a correct output despite fundamentally misunderstanding the task.

**Incorrect + High Understanding** (example, score 8): *"The model correctly identified the intended transformation rule and described it in concept-appropriate terms, but introduced an off-by-one execution error when applying the rule to the test grid."*

**Incorrect + Low Understanding** (example, score 2): *"The reasoning never aligns with the concept definition and instead proposes ad-hoc recoloring steps that do not generalize across the training examples."*

## 4 DISCUSSION

These quadrant cases highlight why accuracy alone is insufficient: it can overestimate reasoning ability (correct-but-flawed) and underestimate it (incorrect-but-sound). Understanding scores help isolate whether errors are conceptual (wrong rule) or procedural (execution mistakes).

We also emphasize a broader interpretability caveat: ConceptARC's concept taxonomy is human-designed, and our judge rubric operationalizes "understanding" as alignment with those human-defined categories. An LLM may arrive at correct solutions via different internal abstractions that do not cleanly map onto the benchmark's concept names, especially if its latent representations carve up the space in a non-anthropocentric way. As a result, low concept-alignment scores should be interpreted as misalignment with ConceptARC's definitions, not necessarily an absence of useful underlying structure.

We also emphasize a broader interpretability caveat: ConceptARC's concept taxonomy is human-designed, and our judge rubric operationalizes "understanding" as alignment with those human-defined categories. An LLM may arrive at correct solutions via different internal abstractions that do not cleanly map onto the benchmark's concept names, especially if its latent representations carve up the space in a non-anthropocentric way. As a result, low concept-alignment scores should be interpreted as misalignment with ConceptARC's definitions, not necessarily an absence of useful underlying structure.

## 5 CONCLUSION

We presented a two-phase evaluation pipeline that disentangles conceptual understanding from output correctness on ConceptARC. Our results show a substantial understanding–correctness gap, suggesting that future benchmarking should report both axes to better diagnose model capabilities.

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

## A  APPENDIX

Supplementary material will be added in the final version.

