# OpenReview forum: "CORRECTNESS WITHOUT CONCEPTS? EVALUATING LLM REASONING BEYOND ANTHROPOCENTRIC TAXONOMIES"
_mathai.club/MathAI/2026/Conference — Submitted to 2026_

### Official Review · Reviewer_nPWh · 2026-03-12
**Correctness Without Concepts? Evaluating LLM Reasoning Beyond Anthropocentric Taxonomies**

**Rating:** 3
**Confidence:** 3

**Review:**

An empirical two-phase evaluation framework for LLM reasoning on ConceptARC; the approach is straightforward but the paper is incomplete, weakly grounded in prior work, and its relevance to MathAI is not established.
The paper proposes a two-phase evaluation pipeline: a solver model (DeepSeek-V3.2) generates reasoning traces on ConceptARC tasks, and a separate judge model (Claude Haiku 4.5) scores the traces for conceptual understanding independently of output correctness. The main finding is a 38% mismatch rate between understanding and correctness across 480 evaluations.

Strengths:
- The core observation — that output correctness and reasoning quality are dissociable — is a valid and practically relevant point for LLM evaluation.
- The quadrant analysis (correct/incorrect × high/low understanding) provides a useful diagnostic framing.
- The reported effect size for spatial reasoning weakness (Cohen's d = 1.53) is concrete and interpretable.

Weaknesses:
1. **Relevance to MathAI.** The paper does not establish a connection between the proposed framework and mathematical reasoning or AI for mathematics. While abstract inductive reasoning over structured patterns may have mathematical components, the authors make no such argument — there is no discussion of formal reasoning, proof structures, mathematical benchmarks, or applications to mathematical problem solving. The submission reads as a general LLM evaluation paper, and its relevance to the MathAI venue should be explicitly motivated.

2. **Incomplete submission.** The appendix, which is referenced multiple times in the main text (including the scoring prompt and concept list), contains only the placeholder "Supplementary material will be added in the final version." This makes it impossible to reproduce or properly evaluate the methodology. Reviewing an incomplete paper is not feasible.

3. **Circular evaluation.** The judge is itself an LLM (Claude Haiku 4.5). The paper does not validate the judge's scores against human annotations or any ground-truth measure of understanding. The central claim — that the judge measures conceptual understanding independently — rests entirely on prompt engineering and is not substantiated. This is a significant methodological gap.

4. **Bibliography.** The paper cites only two works. There is a substantial literature on LLM evaluation, reasoning benchmarks, and the correctness-vs-understanding distinction that is entirely absent. The novelty of the contribution cannot be assessed without this context.

5. **Scale and generalizability.** 480 evaluations across 160 tasks and a single solver model is a limited basis for the broad claims made. Results may be specific to DeepSeek-V3.2 and ConceptARC's particular task format.

6. **The "anthropocentric" framing** introduced in the title and abstract is philosophically interesting but remains underdeveloped. It is raised as a caveat but not operationalized or investigated, and adds little to the technical contribution as presented.

Conclusion
The paper addresses a legitimate problem in LLM evaluation but is submitted in an incomplete state, lacks grounding in prior work, and does not establish relevance to mathematical AI. In its current form it does not meet the standards for publication at MathAI 2026.

---

### Official Review · Reviewer_S61Z · 2026-03-13
**Interesting question, but the central metric is not validated and the submission is incomplete**

**Rating:** 4
**Confidence:** 3

**Review:**

This paper asks a worthwhile question: does output correctness really tell us whether an LLM understood the underlying concept? The basic intuition is good, and the proposed two-phase setup is easy to understand: one model solves ConceptARC tasks, and a second model scores the reasoning trace for “conceptual understanding” while being blind to the final answer. That framing is potentially useful as a diagnostic tool.

However, I do not think the current version is strong enough for acceptance. The main issue is not the intuition, but the construct validity of the measurement. The paper’s central contribution depends on the claim that the judge score captures conceptual understanding independently of correctness. But this score is produced entirely by another LLM, with no human validation, no inter-rater comparison, no calibration study, and no evidence that it tracks “understanding” rather than explanation quality, benchmark vocabulary alignment, or stylistic coherence. As written, the paper shows that one LLM judge can disagree with answer correctness when reading another LLM’s reasoning trace. That is interesting, but it is weaker than the paper’s stated claim.

This concern is made sharper by the paper’s own framing. The title and abstract emphasize reasoning beyond anthropocentric taxonomies, but the method is explicitly built around human-defined ConceptARC concepts and a rubric that rewards concept articulation and concept-rule alignment. In other words, the evaluation is not really beyond anthropocentric taxonomy; it is deeply tied to it. The paper does acknowledge this as a caveat, but that caveat does not just qualify the result — it weakens the interpretation of the main metric. A low “understanding” score may reflect mismatch with the benchmark taxonomy, not necessarily lack of useful conceptual structure.

I also found the submission incomplete in ways that materially affect reviewability. The paper refers to `Appendix ??` for the concept list and the full scoring prompts, but the appendix contains only the sentence “Supplementary material will be added in the final version.” For this paper, those materials are not optional supplement; they are core methodological artifacts. Without them, it is impossible to properly assess or reproduce the judging procedure.

The empirical section is also thinner than the abstract suggests. The paper highlights a 38% mismatch rate, a spatial reasoning weakness with \(d = 1.53\), 34% inconsistency across attempts, and longer traces correlating with failure, but these analyses are not really developed in the body. The study also uses a narrow setup: one solver model, one judge model, one benchmark family, and repeated passes on the same 160 tasks. That is enough for an exploratory pilot, but not enough to support broader claims without much stronger validation. In particular, `160 tasks × 3 passes = 480 evaluations` should not be treated as 480 fully independent observations.

Another weakness is the lack of grounding in prior work. The bibliography contains only two references, which is far too thin for a paper about reasoning evaluation, LLM-as-a-judge methodology, and the distinction between correctness and understanding. It is difficult to assess the novelty of the contribution when the paper is so lightly situated in the existing literature.

Finally, the relevance to MathAI is not convincingly established. ConceptARC involves abstract pattern induction and structured transformations, so there is some possible connection, but the paper does not really develop it. As submitted, this reads more like a general LLM evaluation paper than a mathematical AI paper.

**Strengths**
- The paper raises a real and important evaluation question.
- The correctness-versus-understanding distinction is a useful framing.
- The quadrant analysis is a sensible diagnostic lens.
- The paper includes an intellectually honest caveat about anthropocentric benchmark taxonomies.

**Weaknesses**
- The central “understanding” metric is not validated.
- The evaluation is highly circular because it relies entirely on an LLM judge.
- The appendix is effectively missing even though it contains core methodological details.
- The experimental setup is narrow and the statistical interpretation is weak.
- The paper is weakly grounded in prior work and does not clearly establish venue relevance.

**Suggestions for improvement**
- Validate the judge against human annotations or at least provide agreement analysis across multiple judges.
- Include the full prompts, rubric, and concept list in the submission.
- Clarify exactly what the judge sees and why that is sufficient to assess conceptual understanding.
- Provide threshold sensitivity analysis for the score.
- Expand related work substantially.
- Either motivate the MathAI connection more clearly or test the method on explicitly mathematical reasoning tasks.

**Overall assessment**
The paper raises a meaningful question, and I think there may be a useful paper here. But the current version is too incomplete and too weakly validated for me to trust its central metric. Right now, the contribution depends almost entirely on an unvalidated LLM-as-a-judge setup whose prompts are not even included in the paper. That is not enough for acceptance.

---

### Decision · Program_Chairs · 2026-03-14

**Decision:**

Reject

**Comment:**

After careful evaluation by the Program Committee, we regret to inform you that your submission has not been accepted for presentation at MathAI 2026.

All submissions underwent a rigorous two-stage review process. Unfortunately, the reviewers identified one or more of the following concerns with your paper:

- Insufficient mathematical rigor or novelty relative to the existing body of work in the field;
- Presentation of results that substantially overlap with or rephrase previously published findings without clear original contribution;
- Significant issues with technical quality, including but not limited to broken or non-existent references, unsupported claims, or methodological gaps;
- Indications that the manuscript may have been generated with the assistance of large language models without substantial original intellectual contribution by the authors.

We received a large number of submissions this year, and the selection process was highly competitive. We encourage you to carefully consider the reviewers’ feedback (available through OpenReview), revise your work accordingly, and consider submitting an improved version to a future edition of MathAI or to another appropriate venue.

We appreciate your interest in MathAI and hope you will continue to engage with the conference community.

With kind regards,

MathAI 2026 Program Committee
URL: https://mathai.club
Telegram: https://t.me/MathAI_club
Email: mathai.club@yandex.ru